# CD4+ and CD8+ Circulating Memory T Cells Are Crucial in the Protection Induced by Vaccination with *Salmonella* Typhi Porins

**DOI:** 10.3390/microorganisms9040770

**Published:** 2021-04-07

**Authors:** Luis Ontiveros-Padilla, Alberto García-Lozano, Araceli Tepale-Segura, Tania Rivera-Hernández, Rodolfo Pastelin-Palacios, Armando Isibasi, Lourdes A. Arriaga-Pizano, Laura C. Bonifaz, Constantino López-Macías

**Affiliations:** 1Unidad de Investigación Médica en Inmunoquímica, Hospital de Especialidades del Centro Médico Nacional “Siglo XXI”, Instituto Mexicano del Seguro Social (IMSS), Cuauhtemoc, Ciudad de Mexico 06720, Mexico; luisalberto.ontiveros.p@gmail.com (L.O.-P.); jb_justdoit@hotmail.com (A.G.-L.); tepale.araceli@gmail.com (A.T.-S.); tania.riverahernandez@gmail.com (T.R.-H.); isibasi@prodigy.net.mx (A.I.); landapi@hotmail.com (L.A.A.-P.); 2Facultad de Química, Universidad Nacional Autónoma de México (UNAM), Coyoacan, Ciudad de Mexico 04510, Mexico; rodolfop@unam.mx; 3Departamento de Inmunología, Escuela Nacional de Ciencias Biológicas, Instituto Politécnico Nacional (IPN), Miguel Hidalgo, Ciudad de Mexico 11340, Mexico; 4Consejo Nacional de Ciencia y Tecnología (CONACYT), Benito Juárez, Ciudad de Mexico 03940, Mexico

**Keywords:** *S*. Typhi porins, vaccination, circulating memory, T cells

## Abstract

*Salmonella enterica* serovar Typhi (*S*. Typhi) porins, OmpC and OmpF, are potent inducers of the immune response against *S*. Typhi in mice and humans. Vaccination with porins induces the protection against 500 LD_50_ of *S*. Typhi, life-lasting bactericidal antibodies and effector T cell responses in mice; however, the nature of the memory T cell compartment and its contribution to protection remains unknown. In this work, we firstly observed that vaccination with porins induces in situ (skin) CD4+ and CD8+ T cell responses. Analysis of the porin-specific functional responses of skin CD4+ and CD8+ T cells showed IFN-gamma- and IL-17-producing cells in both T cell populations. The memory phenotype of porin-specific T cells indicated the presence of resident and effector memory phenotypes in the skin, and a central memory phenotype in the skin-draining lymph node. In addition, we demonstrated that vaccination with porins via skin reduces the bacterial burden following challenge. Finally, evaluating the role of the circulating T cell memory population in protection, we showed that circulating memory CD4+ and CD8+ T cells are crucial in porin-mediated protection against *S*. Typhi. Overall, this study highlights the importance of inducing circulating memory T cell responses in order to achieve the optimal protection provided by porins, showing a mechanism that could be sought in the rational development of vaccines.

## 1. Introduction

*Salmonella enterica* (*S. enterica*) is a human bacterial pathogen that represents a serious public health problem in low- and middle-income countries. The World Health Organization has declared *S. enterica* as a high priority pathogen for the development of new antibiotics and vaccines due to the emergence of multi-drug resistant strains [1]. *S*. *enterica* serovar Typhi (*S*. Typhi) is the cause of typhoid fever, of which approximately 11 million cases and 117,000 deaths are reported annually [2]. Vaccine development against *S*. *enterica* continues to be a challenge; only three vaccines against typhoid fever are currently licensed, some of them have limitations of effectiveness in young children, and none of them are able to provide cross-protection against non-typhoidal strains, which cause more than 93 million infections annually [3].

Previous work from our laboratory showed that the antibody response in typhoid fever patients strongly reacted to *S*. Typhi OmpC and OmpF porins [4]. Porins were purified from *S*. Typhi and administered in mice, providing protection against 500 lethal doses of *S*. Typhi [5]. In search of the mechanisms of protection induced by immunization with purified porins, previous studies described a strong and life-lasting bactericidal antibody response in mice [6]. In addition, it was observed that repeated vaccination with *S*. Typhi porin programs type I follicular helper T cell responses contribute to the diversification of B cell memory and promote a protective IgM antibody response; additionally, the presence of porin-specific IgM and IgG antibody responses eleven years after a single immunization with 10 µg of porins in human volunteers has been reported [7]. This highly immunogenic response can be attributed to the intrinsic adjuvant capacity of porins, when they are co-immunized with T-dependent and T-independent antigens, the antigen-specific T cell and antibody responses are found to be increased [8].

It has been reported that porins are a key target of T cell responses triggered by immunization with Ty21a typhoid vaccine [9,10], and specific porin-epitopes responsible of such T cell responses have been identified [11,12]. Following immunization with porins, mice develop proliferative and effector responses against OmpC and OmpF [13]. Likewise, in a phase 1 clinical trial, vaccination with porins in human volunteers showed the induction of antigen-specific cellular responses [10,14]. However, the generation of a memory T cell compartment and its contribution in the protection induced by *S.* Typhi porins has not been explored.

The immunization of antigens and vaccines into the skin has been widely used as a model to study memory T cell responses upon vaccination. This vaccination route has proven useful to evidence the presence of different memory T cell subsets: (i) resident memory T cells (CD44+, CD62L−, CD103+); (ii) effector memory T cells (CD44+, CD62L−, CD103−); and (iii) central memory T cells (CD44+, CD62L+, CD103−) T cells [15,16,17]. The main difference between these subsets is that effector and central memory T cells are able to recirculate systematically without the capacity to reside in tissues, while the opposite applies to resident T cells [18,19]; furthermore, the resident T cell subset is characterized by being located in tissues difficult to reach by circulating T cells, such as intestinal or lung mucosa [20].

Both resident and circulating memory T cells have been shown to display a key role in vaccine and infection-mediated protection against viral and bacterial pathogens, such as influenza virus, *Yersinia* sp. and live attenuated *Salmonella* sp. [21,22,23]. In this work, we propose to investigate the specific memory T cell compartment, with particular interest in the circulating and resident memory T cell subsets induced by porin vaccination and their potential role in protection against *S*. Typhi.

## 2. Materials and Methods

### 2.1. Animals

Male C57BL/6J mice aged 6–8 weeks old were purchased from Bioinvert^®^ trading house (Mexico City, Mexico) and kept for the duration of the experiments at the animal facility of the Unidad de Medicina Experimental, Facultad de Medicina de la UNAM, with continuous cleaning and constant supply of food and water. At the end of the experiments, mice were sacrificed according to the official national regulations (NOM-062-ZOO-1999).

### 2.2. Bacteria

Live *S*. Typhi ATCC 9993 strains were obtained from ATCC (Manassas, VA, USA) and grown in LB broth (Sigma cat. L3397, St. Louis, MO, USA) at 37 °C under stirring for approximately 16 h until reaching a bacterial concentration of 10^9^ CFU/mL. Aliquots of 10^7^ live *S*. Typhi were used for the challenge or were heated at 60 °C for 2 h to obtain heat-inactivated *S*. Typhi (HIS), which was immunized intraperitoneally as a positive control.

### 2.3. S. Typhi Porins

*S*. Typhi porins were purified and produced using previously published methods [14] (4) from the *Salmonella* Typhi ATCC 9993 strain, at Unidad de Investigación Médica en Inmunoquímica, Hospital de Especialidades del Centro Médico Nacional “Siglo XXI”, Instituto Mexicano del Seguro Social, in Mexico City. Briefly, *S*. Typhi was grown in glucose-supplemented minimal A medium and porins were extracted from bacteria using sodium dodecyl sulphate buffer. Proteins were purified by molecular exclusion chromatography using a Sephacryl S-200 column and purification was corroborated by SDS-page gel electrophoresis. Lipopolysaccharide content was evaluated using a Limulus amoebocyte lysate (LAL) assay (Endosafe KTA, Charles River Laboratories, Wilmington, MA, USA), and the batch used in the study contained no detected lipopolysaccharide present (detection limit, 0.2 ng LPS/mg protein).

Proteinase K-digested porins (PorK) were obtained by digesting porins in proteinase K (New England Biolabs cat. P8102, Ipswich, MA, USA) (1 µg/500 µg of porins) at 37 °C for 2 h and then heated at 60 °C for 10 min. PorK was used in the experiments as a negative control to evaluate the effect of the potentially remaining lipopolysaccharide that could be associated to the porins structure and in quantities below the LAL assay detection limit. Vehicle (Veh) was used as negative control in the experiments and consisted of Tris base (Bio-rad cat. 1610716, Hercules, CA, USA) buffer 1.0 M pH 7.4 in which porins were injected into the mice.

### 2.4. Vaccination

PorID mice were injected intradermally in both ears with 5 µg of porins in a volume of 10 µL of vehicle per ear (total dose of 10 µg of porins); PorIP mice were injected intraperitoneally with 10 µg of porins in a volume of 200 µL of vehicle. Negative control groups were injected with 10 µL of vehicle or 10 µg of PorK. Positive control mice were injected with 10^7^ CFU of HIS intraperitoneally. For the evaluation of cellular and memory T cell responses, all groups were boosted with 10 µg of porins intradermally (Figure 1, Figure 2 and Figure 3). For the protection assays (Figure 4 and Figure 5) all groups received a homologous boost. All boosts were carried out at day 14 post-injection.

### 2.5. Delayed-Type Hypersensivity (DTH)

Ear thickness was measured with a micrometer (Truper cat. 14401, Mexico City, Mexico) in a delayed-type hypersensitivity test at 24 and 48 h following PorID boost.

### 2.6. T Cell Extraction and Stimulation

Cells were extracted from the ear skin and skin dLN of injected mice by macerating the tissue followed by culture cell suspensions in supplemented RPMI (Gibco cat. 1640, Waltham, MA, USA) culture media at 37 °C for 24 h. After incubation, a cell pellet was obtained by centrifugation at 2500 rpm for 5 min and erythrocyte lysis was carried out with NH4Cl 0.15 M. Cell viability was measured using trypan blue staining and cells were counted in a Neubauer chamber. A total of 10^6^ live cells extracted from skin tissue and skin dLN were stimulated with the previously described peptides OmpC_241–255_ (0.5 µg) and OmpF_201–215_ (0.5 µg) [7] at 37 °C for 5 h.

### 2.7. T Cell Immunophenotyping

Following stimulation, extracellular staining was carried out with anti-mouse TCR-β (Biolegend cat. 109228, San Diego, CA, USA), CD4 (Biolegend cat. 100414, San Diego, CA, USA), CD8-α (Biolegend cat. 100725, USA), CD44 (BD cat. 563114, Franklin Lakes, NJ, USA), CD62L (Biolegend cat. 104408, San Diego, CA, USA) and CD103 (Biolegend cat. 121426, San Diego, CA, USA) antibodies. Intracellular staining was carried out using Fix and perm solutions (BD cat. 554714, Franklin Lakes, NJ, USA) and staining with anti-mouse IFN-γ (Biolegend cat. 505806, San Diego, CA, USA) and IL-17 (Biolegend cat. 506916, San Diego, CA, USA) antibodies. Stained cells were acquired in a BD Facs Canto II flow cytometer (BD, Franklin Lakes, NJ, USA) from the Instrument center at Hospital de Especialidades del Centro Médico Nacional “Siglo XXI”, Instituto Mexicano del Seguro Social.

### 2.8. IgG and IgM Antibody Titers

Mice were bled at day 28 following the first immunization, blood was centrifugated at 2500 rpm/10 min, and serum was collected. We evaluated porin-specific antibody titers by ELISA, coating polystyrene 96-well plates (Corning cat. 3590, Corning, NY, USA) with 1 µg of porin in 100 µL of carbonate buffer (pH = 9.6), serum was diluted along the plate, and porin-specific antibodies were detected with anti-mouse IgM (Invitrogen cat. 62-6820, Waltham, MA, USA) or IgG secondary antibody (Invitrogen cat. 62-6520, Waltham, MA, USA). Plates were revealed with 10 µg of *o*-phenylenediamine (Thermo Scientific cat. 34005, Waltham, MA, USA) and 10 µL of hydrogen peroxide (Sigma cat. H1009, St. Louis, MO, USA) in 100 µL of citrate buffer (pH = 4.5), and well absorbances were measured at 492 nm in a plate reader (Thermofisher Multiskan 355, Waltham, MA, USA).

### 2.9. CD4 and CD8 Depletion

GK1.5 and TIB-210 depleting monoclonal antibodies, as well as the non-specific isotype control III-10 monoclonal antibody, were produced in-house. At days 25, 28 and 31 following vaccination with porins, mice were injected intraperitoneally with monoclonal antibodies GK1.5 (400 µg/day) and TIB-210 (200 µg/day) in order to deplete CD4+ and CD8+ cells, respectively, or with monoclonal antibody III-10 (400 µg/day) as an isotype control.

### 2.10. Challenge with Bacteria and CFU Count

Mice were challenged intraperitoneally with 10^5^ (Figure 4) or 10^8^ (Figure 5) CFU of *S*. Typhi 28 days after the first vaccination. One day (Figure 4) or five days (Figure 5) post-challenge, spleen and liver from mice were macerated in sterile PBS 1X. Concentrated mash in 1:10 and 1:100 dilutions were cultivated on LB (Sigma cat. L2897, St. Louis, MO, USA) medium plates and incubated at 37 °C for 14 h. Following incubation, CFUs were counted to determine CFU/g of organ for each sample.

### 2.11. Survival Test and Clinical Score

Survival percentage was measured following mice from all groups for 5 days post-challenge. During the same time, a clinical score was given by observing bristling, peritoneal hyperplasia, wounds, immobility, and other clinical signs, awarding a 1–5 score to each group of mice according to a previously described scale for assessing pain and distress in mice [24].

## 3. Results

### 3.1. Vaccination with S. Typhi Porins Induces CD4+ and CD8+ T Cell Responses

In order to test whether vaccination with porins could induce T cell responses that would allow us to search different memory T cell subsets, we immunized mice using a prime-boost strategy followed by a delayed-type hypersensitivity (DTH) test. To evidence the presence of a local cellular response against porins, DTH tests were carried out in a group of mice that were primed and boosted with porins intradermally (PorID). On the other hand, another group was primed with porins intraperitoneally (PorIP) and boosted with PorID, where a positive dermal DTH response would indicate that the priming with PorIP created a systemic cellular response that reached skin and was later expanded following the local boost. All experimental groups were boosted with PorID (Figure 1A). The DTH test showed a sustained cellular response in both groups, denoted by a persistent increase in skin thickness at 24 and 48 h post-boost in the vaccinated site (Figure 1B). 

The positive DTH response in porin-immunized mice suggested the presence of local and systemic cellular responses regardless of the immunization route used for priming. We then wanted to corroborate that the DTH response after vaccination was a consequence of an increase in the number of T cells at the immunization site. Using flow cytometry, we found larger populations of CD4+ and CD8+ T cells in the skin of porin-vaccinated mice, especially in the PorID group (Figure 1C). Cell quantifications showed that CD4+ T cell percentages (Figure A1A) and total numbers (Figure 1D) in the skin of vaccinated mice were three-fold higher compared to the negative control groups. In the case of skin CD8+ T cells, although the percentage was found at the same levels as our negative control groups (Figure A1B), we found an increment in the total number of this cells (Figure 1E). To test whether the positive DTH response and the increase in CD4+ and CD8+ T cells was related to the presence of local and systemic (circulating) memory T cells, we then characterized the memory phenotype, and the functionality of the T cells present in the skin.

### 3.2. Vaccination with S. Typhi Porins Induces Functional Resident and Effector Memory CD4+ and CD8+ T Cell Responses

We first assessed the specificity, and the functionality of the skin T cells through the expression of IFN-gamma (IFN-γ) and IL-17, cytokines produced as a consequence of a porin-specific recall response [7,8]. The specific T cell cytokine profile in the vaccinated site revealed the expression of IFN-γ and IL-17 by CD4+ T cells after the stimulation with the peptides OmpC_241–255_ and OmpF_201–215_ (Figure 2A). In addition, the percentage (Figure A2A) and total cell number (Figure 2B) of IFN-γ-positive CD4+ T cells represented a seven-fold increase in the vaccinated mice compared with controls. Similarly, the percentage (Figure A2B) and total cell number (Figure 2C) of IL-17-positive CD4+ T cells represented a five-fold increase in the vaccinated mice compared with controls. The presence of functional CD8+ T cells in the skin was also evident; we were able to measure an increase in percentage (Figure A2C) and total cell number (Figure 2D) of IFN-γ-producing CD8+ T cells. On the other hand, although the percentage of IL-17 expressing CD8+ T cells did not have significant differences compared with the negative controls (Figure A2D), the total number of CD8+ T cells showed a significant increase in comparison with the negative controls (Figure 2E).

To evaluate if porin-specific T cells found in the skin of vaccinated mice had a T cell memory-like phenotype, we analyzed the expression of CD62L and CD103 markers in IL-17 and IFN-γ and skin CD44+ T cells (Figure 2F). We were able to observe the presence of porin-specific resident memory (Figure 2G,H) and effector memory (Figure 2I,J) CD4+ and CD8+ T cell populations in the skin of vaccinated mice, while these memory T cell subsets were almost absent in negative control groups. The number of central memory CD4+ or CD8+ T cells in skin was not found to be increased (counting graphics not shown) in vaccinated groups compared to controls; however, because central memory T cells are mostly located in lymphoid organs, their presence in the skin-draining lymph node (dLN) was further investigated. 

### 3.3. Vaccination with S. Typhi Porins Induces Functional Central Memory CD4+ and CD8+ T Cell Responses

To evaluate the presence of central memory T cells, we obtained cells from the skin dLN and carried out the same data analysis as for skin T cells. We firstly analyzed the presence of CD4+ and CD8+ T cells in the skin dLN (Figure 3A) and found an increase in the total number of CD4+ T cells (Figure 3B), but not in the total number of CD8+ T cells (Figure 3C) in samples from porin-vaccinated mice compared with negative control groups. Following the same strategy used to analyze skin cells, we then evaluated the functionality and specificity of skin dLN T cells by analyzing the expression of IFN-γ and IL-17 (Figure 3D). A significant increment in the percentage (Figure A3A) and in the total number (Figure 3E) of IFN-γ-producing CD4+ T cells was observed, while a similar increment was found in the percentage (Figure A3B) and in the total number (Figure 3F) of IL-17-producing CD4+ T cells. Contrary to what was observed in the skin, the presence of functional CD8+ T cells was clearly evident, the percentage (Figure A3C) and the total number (Figure 3G) of IFN-γ-producing CD8+ T cells were significantly higher compared to controls; a similar finding was observed in the percentage (Figure A3D) and the total number (Figure 3H) of IL-17-producing CD8+ T cells.

Finally, we evaluated the memory phenotype of specific T cells in the skin dLN (Figure 3I). We were able to observe a predominant presence of central memory CD4+ (Figure 3J) and CD8+ (Figure 3K) T cells, while those memory T cells were practically absent in negative control groups. On the other hand, resident and effector memory T cells were not found to be increased for any of the T cell populations in the skin dLN (counting graphics not shown) for any of the experimental groups. Up to this point, we were able to detect the presence of resident and circulating memory T cell subsets following vaccination with porins; we next sought to evaluate their role in protection against *S*. Typhi challenge.

### 3.4. Vaccination with S. Typhi Porins Reduces the Bacterial Burden Following Challenge

Porins have been demonstrated to be protective when delivered via the intraperitoneal route, but in this study, to evaluate the role of the memory T cell subsets in protection, it was important to determine if administrating porins via the skin was also able to induce protection and if it could be as or more immunogenic than the intraperitoneal route. Skin vaccination with porins in a prime-boost scheme (Figure 4A) was able to induce porin-specific IgM (Figure 4B) and IgG (Figure 4C) titers at similar levels to PorIP or HIS, and significantly higher levels compared to negative controls. We then evaluated if both routes of vaccination (PorID or PorIP) were able to control the bacterial burden in the spleen (Figure 4D) and liver (Figure 4E), observing that both vaccination routes with porins leave the organs of most porin-vaccinated mice with CFU levels below our limit of detection, similar to levels observed in HIS-immunized mice. Thus, we confirmed that the vaccination routes that induce memory T cell subsets are able to control the *S*. Typhi burden after challenge. 

### 3.5. Circulating Memory CD4+ and CD8+ T Cells are Crucial in the Protection Induced by S. Typhi Porins

Next, we wanted to evaluate whether circulating and resident memory T cells have a relevant contribution to the protection induced by vaccination with porins. We depleted CD4+ and CD8+ T cells in porin-vaccinated mice with the intraperitoneal administration of anti-CD4 and anti-CD8 monoclonal antibodies at day 28 post-vaccination (Figure 5A). Following the depletion of CD4+ and CD8+ T cells, we evaluated whether circulating and/or resident T cells were depleted. We were able to confirm the depletion of circulating memory T cells, while skin T cells could not be depleted (Figure 5B), which was corroborated in the total count of CD4+ and CD8+ T cells in skin and blood (Figure A4).

Subsequently, we evaluated if the absence of circulating memory CD4+ and CD8+ T cells following vaccination with porins has an impact in the protection against bacteria. The CD4- and CD8-depleted mice were challenged with a lethal dose of *S.* Typhi after being immunized and once porin-specific antibody titers were measured. Firstly, we observed that IgM (Figure 5C) and IgG (Figure 5D) antibody responses were not affected by the depletion of both circulating memory CD4+ or CD8+ T cell populations, confirming that we were able to evaluate the specific role of the lack of cellular responses in the control of bacterial dissemination. We then found that the absence of circulating memory CD4+ and CD8+ T cells resulted in a significant increase in the average bacterial burden in spleen compared to controls (Figure 5E). A similar effect was observed in the liver, where the absence of circulating memory CD4+ and CD8+ T cells increased the number of bacteria found in this organ (Figure 5F).

Finally, we report that the lack of CD4+ and CD8+ circulating memory T cells in porin-vaccinated mice had a major detrimental effect in protection against challenge. We observed a decrease of 50% and 80% in survival for depleted CD4+ and CD8+ mice, respectively, in comparison to non-depleted vaccinated mice (Figure 5G). The clinical scores assigned during the course of challenge also showed that the absence of CD4+ and CD8+ circulating T cells had a negative impact in the severity of symptoms, with depleted mice having higher clinical scores compared to non-depleted vaccinated mice (Figure 5H). These final results provide evidence of the crucial role of the circulating (effector and central) memory CD4+ and CD8+ T cells in the protection against *S*. Typhi induced by porins.

## 4. Discussion

The porin-induced mechanisms involved in protection against *S.* Typhi infection are not yet fully understood, although a robust antibody response and the presence of effector T cell responses induced by vaccination have been reported to play a key role. However, the porin-induced memory T cell compartment has not been explored, and it could represent an important compartment in the protection against *S.* Typhi infection. Therefore, in this work we evaluated the induction of memory T cell subsets by vaccination with porins, and the role of circulating memory T cells in the porin-mediated protection against *S.* Typhi challenge.

Here, we showed that *S.* Typhi porins induced a specific DTH response following a prime-boost vaccination regime. *S.* Typhi outer membrane proteins were used in DTH tests, but as only as adjuvants when co-immunized with Mycobacterium sp. [25]. We also report an increase in the number of CD4+ and CD8+ T cells in the vaccinated site with porins, which is in agreement with previous reports that correlate a positive DTH test with an increase in T cells in the DTH tested site [26].

We then observed the presence of functional and specific CD4+ and CD8+ T cells in skin through the expression of cytokines after porin-specific stimulation. It is important to mention that although functional CD4+ T cells were found in the skin and the skin dLN, functional CD8+ T cells were mainly found in the skin dLN and not in the skin. CD8+ T cells in the skin dLN were not found to be increased in porin-vaccinated mice; probably, for this reason, only a small number of functional CD8+ T cells could reach the skin and be detected at the time this site was analyzed. The study of the functional and porin-specific response of the CD4+ and CD8+ T cells in skin showed that both T cells expressed IFN-γ or IL-17 cytokines; however, different patterns of expression of IFN-γ and IL-17 were found depending on the analyzed site. Skin dLN T cells are prone to IFN-γ expression, which is in agreement with previous reports [7]. On the other hand, although a Th1/Th17 profile following vaccination with bacterial proteins has been described [27], we found a larger population expressing IL-17 in skin; this particular pattern can be explained because Th17 cells give rise to long-term resident memory T cells following immunization, which can respond immediately against bacterial pathogens [28].

Vaccination with porins induced the presence of antigen-specific circulating and resident memory T cells. The effector and resident memory CD4+ and CD8+ T cell populations were found in the skin, while the presence of central memory CD4+ and CD8+ T cells was evident in the skin dLN. Induction of the three memory T cell subsets by vaccination with unadjuvanted antigens is not a commonly described phenomenon; it has been reported mostly for adjuvanted antigens, and in those cases the memory T cell response is mainly directed to a circulating (effector and central) memory T cell profile [29]. In this particular case, we demonstrate that immunization with *S.* Typhi porins is not only able to induce the circulating memory subset, but also resident memory T cells, which could be due in part to their intrinsic adjuvanticity. This uncommon feature of porins has triggered interest to use them as adjuvants for other subunit antigens, microorganisms, and potential cancer therapies [8,30,31].

In the case of protection against *S.* Typhi, results from a parabiosis model using mice immunized with the live vaccine strain demonstrated that circulating and non-circulating (resident) T cell subsets are required to achieve the optimal protection against *S.* Typhi challenge [23]. In this finding, although the optimal protection was obtained in mice having both circulating and resident T cells, the presence of only the circulating subset conferred partial protection against *S.* Typhi. For this reason, further studies are required to elucidate the exact role of the circulating T cell subset in the protection against *S.* Typhi. The characterization of the role in protection of memory T cell subsets triggered by vaccination with porins is a crucial step to understand the mechanisms of protection elicited by this particular antigen.

We showed that vaccination with porins induced a strong antibody and protective response, and that the presence of circulating memory CD4+ and CD8+ T cells is required in order to achieve optimal protection against the bacterial challenge. Such immunological features have not been widely reported using purified bacterial antigens, although they are a common finding in studies using whole bacteria or fungi immunization [32,33]. It is important to highlight that, even when not all circulating CD4+ T cells could be depleted using very high doses of depleting antibody, our depletion model allowed us to observe the direct effect of the absence of circulating memory T cells in porin-mediated protection, given that porin-specific IgM and IgG antibodies with potential bactericidal activity remained unchanged at the time of challenge. In addition, the lack of circulating memory T cells was induced only before carrying out the challenge, at the time when we had evidenced the presence of memory circulating T cell subsets in previous experiments.

A more precise evaluation of the contribution of resident T cells in protection remains technically challenging, mainly due to the inaccessibility of depleting antibodies into desired tissues and the difficulties to separate resident T cells from the rest of the immune response in parabiosis or depletion experiments. For these reasons, numerous studies report the protective role of memory resident T cells included within the whole non-circulating compartment of the immune response [21,22,23]. In our particular case, we were able to evaluate the role of the circulating memory subset in the porin-mediated protection, demonstrating a crucial role of these cells in controlling bacterial dissemination. In mice lacking circulating memory T cells, the porin-mediated protection was not completely lost, potentially due to the presence of bactericidal antibodies; however, given that resident T cells were still present, we cannot discard a potential role of this subset in the protection against *S.* Typhi challenge and such role should be further investigated.

Taken together, our results showed that vaccination with *S.* Typhi porins generates specific and functional, circulating and resident memory CD4+ and CD8+ T cell responses, with an outstanding role of the circulating subset in the porin-induced protection. The finding of the importance of circulating memory CD4+ and CD8+ T cell responses in protection adds a piece into the puzzle of porin-induced mechanisms of protection where antibodies are well known to play a key role. We hope that this work contributes to the understanding and dissection of the immune response components involved in porin-mediated protection and in the correlates of protection against *S.* Typhi infection.

## Figures and Tables

**Figure 1 microorganisms-09-00770-f001:**
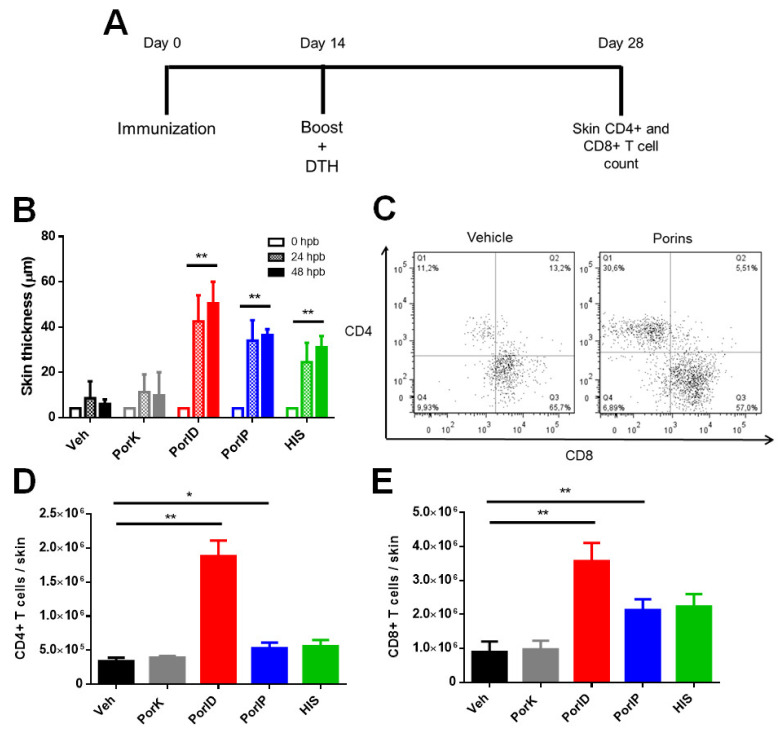
Vaccination with *S*. Typhi porins induces CD4+ and CD8+ T cell responses. C57BL/6J mice were injected intradermally in the ear with vehicle (Veh), proteinase K-digested porins (PorK), porins (PorID), or intraperitoneally with porins (PorIP) or heat-inactivated *S*. Typhi (HIS) at day 0. All groups were boosted at day 14 with porins injected intradermally in the ear (**A**). Skin thickness was measured at 24 and 48 h post-boost (hpb) (**B**). Representative plot (**C**) of the total number of CD4+ (**D**) and CD8+ (**E**) T cells present in the skin at day 28 (*n* = 7; 2 independent experiments; significant difference for a Kruskal–Wallis test ** *p* < 0.05 * *p* < 0.1 with Dunn’s multiple comparison).

**Figure 2 microorganisms-09-00770-f002:**
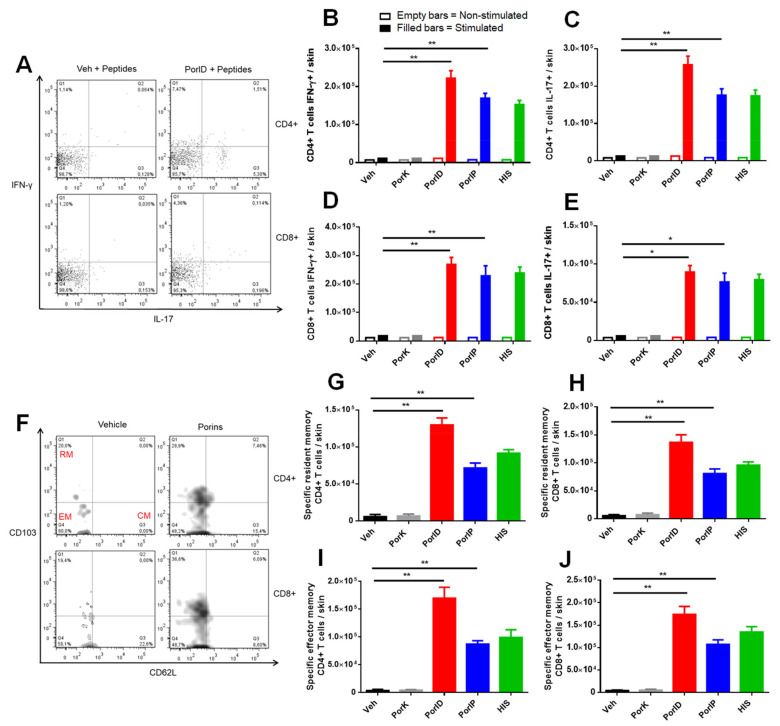
Vaccination with *S*. Typhi porins induces functional resident and effector memory CD4+ and CD8+ T cell responses. C57BL/6J mice were injected intradermally in the ear with vehicle (Veh), proteinase K-digested porins (PorK), porins (PorID), or intraperitoneally with porins (PorIP) or heat-inactivated *S*. Typhi (HIS) at day 0. All groups were boosted at day 14 with porins injected intradermally in the ear. At day 28, expression of IFN-γ and IL-17 was evaluated in skin T cells following stimulation with *S*. Typhi porins peptides. Representative plot (**A**) used to analyze the total number of IFN-γ + (**B**) and IL-17+ (**C**) CD4+ T cells. Total number of IFN-γ + (**D**) and IL-17+ (**E**) CD8+ T cells. At day 28, memory phenotype T cells (CD44+) were evaluated in the skin. Representative plot (**F**) used for the detection of both effector memory (CD62L−, CD103−) (**G**,**H**) and resident memory (CD62L−,CD103+) (**I**, **J**) CD4+ and CD8+ T cell populations, respectively (CM, central memory, EM, effector memory, RM, resident memory) (*n* = 7; 2 independent experiments; significant difference for a Kruskal–Wallis test ** *p* < 0.05 * *p* < 0.1 with Dunn´s multiple comparison).

**Figure 3 microorganisms-09-00770-f003:**
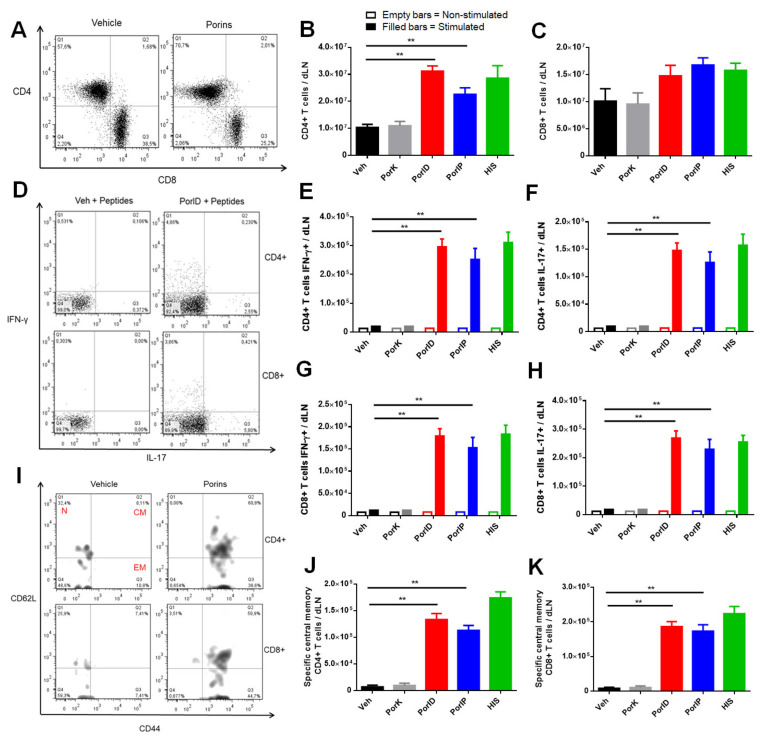
Vaccination with *S*. Typhi porins induces functional central memory CD4+ and CD8+ T cell responses. C57BL/6J mice were injected intradermally in the ear with vehicle (Veh), proteinase K-digested porins (PorK), porins (PorID), or intraperitoneally with porins (PorIP) or heat-inactivated *S*. Typhi (HIS) at day 0. All groups were boosted at day 14 with porins injected intradermally in the ear. T cells from the skin dLN were analyzed at day 28. Representative plot of the count of cell number (**A**) for both CD4+ (**B**) and CD8+ (**C**) populations in the skin dLN. Representative plot used (**D**) to analyze expression of IFN-γ and IL-17 in CD4+ and CD8+ dLN T cells following stimulation with *S*. Typhi porin peptides. Total number of IFN-γ- (**E**) and IL-17- (**F**) producing CD4+ T cells. Total number of IFN-γ- (**G**) and IL-17- (**H**) producing CD8+ T cells. Central memory phenotype (CD44+, CD62L+) was evaluated in skin dLN porin-specific T cells. Representative plot (**I**) used to analyze central memory CD4+ (**J**) and CD8+ (**K**) populations (CM, central memory, EM, effector memory, N, naive) (*n* = 7; 2 independent experiments; significant difference for a Kruskal–Wallis test ** *p* < 0.05 with Dunn’s multiple comparison).

**Figure 4 microorganisms-09-00770-f004:**
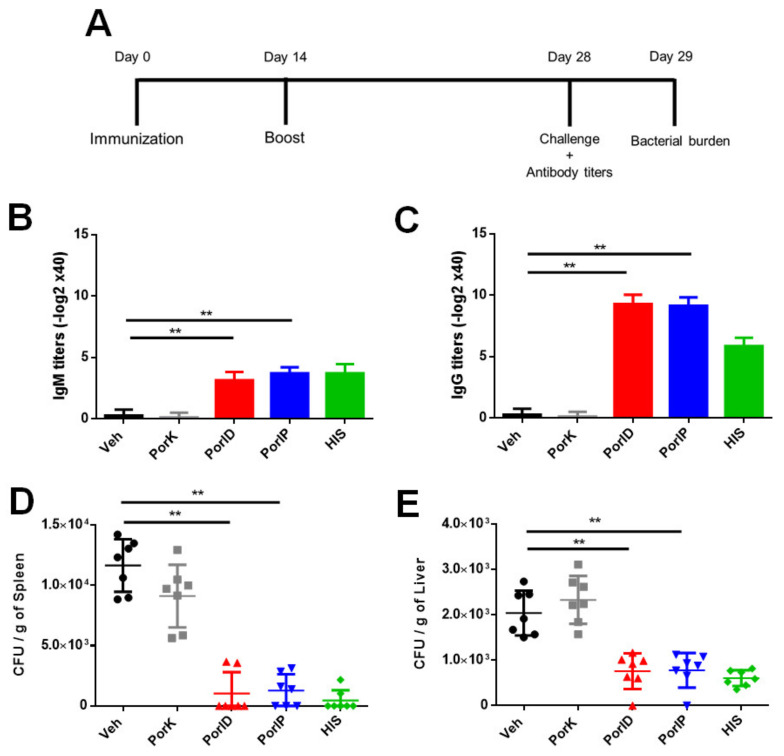
Vaccination with S. Typhi porins reduces the bacterial burden following challenge. C57BL/6J mice were injected intradermally in the ear with vehicle (Veh), proteinase K-digested porins (PorK), porins (PorID), or intraperitoneally with porins (PorIP) or heat-inactivated *S*. Typhi (HIS) at day 0 with a homologous boost at day 14 (**A**). At day 28, IgM (**B**) and IgG (**C**) serum titers were measured, and mice were challenged intraperitoneally with 10^5^
*S*. Typhi CFU. At day 29, bacterial burden in the spleen (**D**) and liver (**E**) was determined (*n* = 7; 3 independent experiments; significant difference for a Kruskal–Wallis test ** *p* < 0.05 with Dunn’s multiple comparison).

**Figure 5 microorganisms-09-00770-f005:**
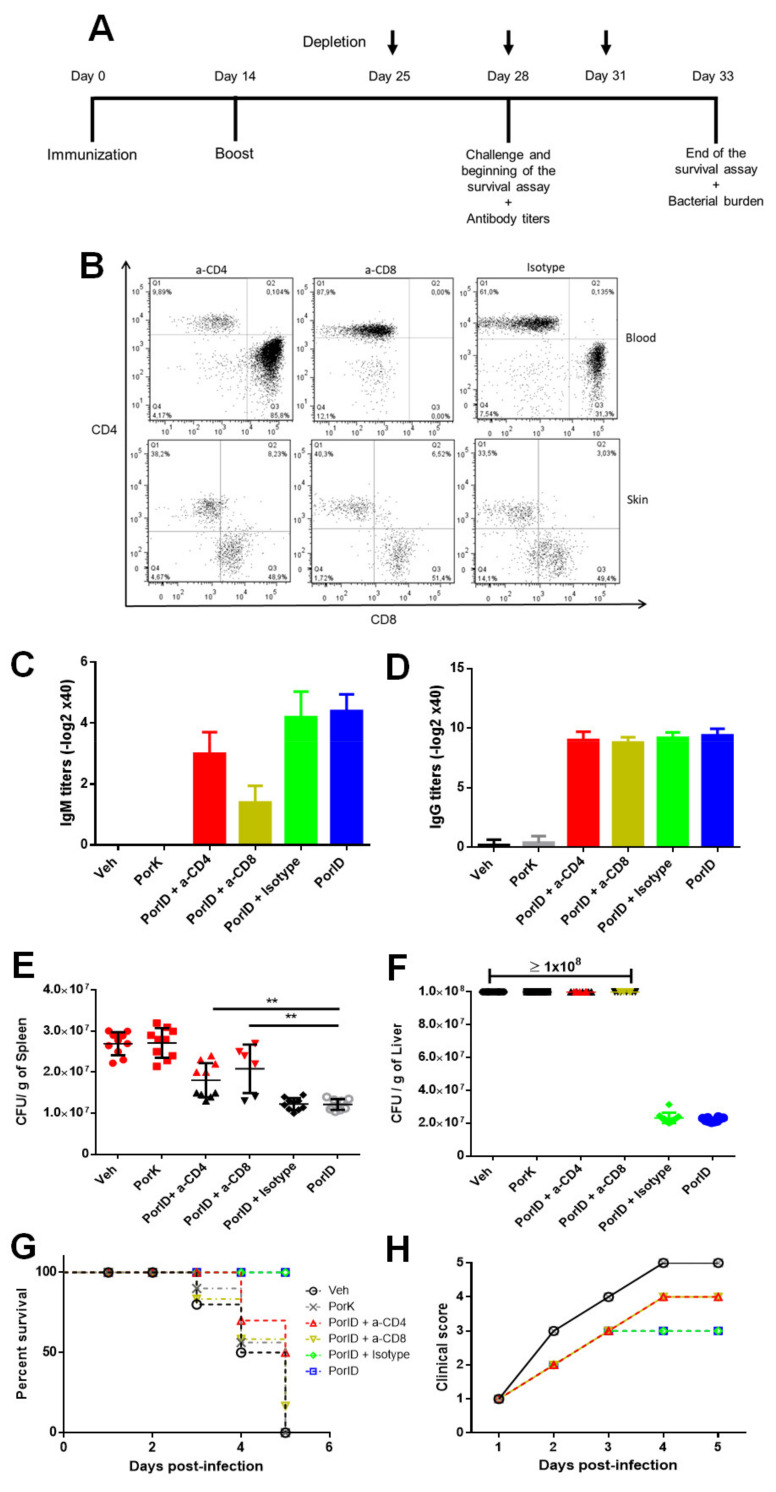
Circulating memory CD4+ and CD8+ T cells are crucial in the protection induced by *S*. Typhi porins. C57BL/6J mice were injected intradermally in the ear with vehicle (Veh), proteinase K-digested porins (PorK), porins (PorID), or intraperitoneally with porins (PorIP) or heat-inactivated *S.* Typhi (HIS) at day 0 with a homologous boost at day 14. PorID immunized mice were divided into 4 conditions: anti-CD4 (a-CD4), anti-CD8 (a-CD8), isotype control (Isotype) and no treatment (PorID). At days 25, 28 and 31, CD4+ and CD8+ T cells were depleted with GK1.5 and TIB-210 monoclonal antibodies, respectively, in a-CD4 and a-CD8 groups (**A**). Representative dot-plot of the depletion of circulating CD4+ and CD8+ T cells at day 28 (challenge day) (**B**). IgM (**C**) and IgG (**D**) serum titers were measure before challenge at day 28. Mice were challenged with 10^8^
*S*. Typhi CFU and bacterial burden in spleen (**E**) and liver (**F**) was quantified at 5 days post-challenge (mice that succumbed during challenge are marked in black). Mouse survival (**G**) and clinical scores (**H**) were recorded once daily for 5 days post-challenge (*n* = 10; 3 independent experiments; significant difference for a Kruskal–Wallis test ** *p* < 0.05 with Dunn’s multiple comparison).

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
