# Peer review of "CD4+ and CD8+ Circulating Memory T Cells Are Crucial in the Protection Induced by Vaccination with Salmonella Typhi Porins"

_microorganisms, 2021, doi:10.3390/microorganisms9040770_

Round 1
Reviewer 1 Report
Well done and straightforward. A solid follow-up on previous studies from the same group. No major comments.
Author Response
We thank and appreciate the reviewer’s comment.
Reviewer 2 Report
The methodology of the experiment is adequately described, but it would be better to present it by visual means.
The figures are too many and respectively very small, which makes them very difficult to read. I recommend reducing their number and bringing them out as additional material to the publication.
Author Response
We thank the reviewer’s comments. As suggested by the reviewer, we added an schematic view of the experimental strategy in figures 1 and 4. In addition we reduced the number of panels in figures 1, 2 and 3, the removed panels are now as supplementary information as figures A1, A2 and A3.
Reviewer 3 Report
This is an excellent work that is clearly explained, and the conclusions are supported by the data.
My only comment was that I found the materials and methods section to be extensively repeated in the results section. I am not sure how this fits with the Journal’s style but it would not be acceptable in the other journals for which I review.

Author Response
We kindly thank the reviewer’s comment. As per the reviewer’s suggestion, he have reviewed the results section and eliminated all information that was already described in the materials and methods section.